# Economic Rationality in Decision-Making Regarding Newborn Screening: A Case Study in Quebec

**DOI:** 10.3390/ijns10020036

**Published:** 2024-05-13

**Authors:** Van Hoa Ho, Yves Giguère, Daniel Reinharz

**Affiliations:** 1Department of Social and Preventive Medicine, Faculty of Medicine, Laval University, Quebec City, QC G1V 5C3, Canada; van-hoa.ho.1@ulaval.ca; 2Department of Molecular Biology, Medical Biochemistry and Pathology, Faculty of Medicine, Laval University, Quebec City, QC G1V 5C3, Canada; yves.giguere@crchudequebec.ulaval.ca; 3CHU de Québec-Université Laval Research Center, Quebec City, QC G1V 4G2, Canada

**Keywords:** neonatal care, economic evaluation, decision-making, evidence-based medicine, neo-institutional framework

## Abstract

Health systems in high-resource countries recognize the importance of making decisions about the services offered to the population based on scientific evidence. Producing this evidence is especially challenging in areas such as newborn care where the frequency of conditions is rare. However, methodological advances in the field of economic evaluation could change how this evidence is used in decision-making. This study aimed to investigate how decision-makers in the Canadian province of Quebec perceive the value of recent advances in economic evaluations for perinatal studies and how these advances might affect the offer of neonatal interventions in the public health care system. A qualitative study was conducted. A total of 10 policymakers were interviewed. A neo-institutional conceptual framework highlighting three dimensions, structure, power, and interpretive schemes, was used for data collection and analyses. Structural factors, interpretative schemes, and power management between the groups concerned concur to ensure that providing services to newborns is not hindered by the difficulty of producing evidence. They also ensure that the decisions regarding which disease to screen for take into consideration the specificity of neonatology, in particular, the social value given to children not captured by available evidence.

## 1. Introduction

In industrialized countries, the choice of interventions offered in the healthcare system tends to take into account different types of scientific evidence, notably epidemiological and economic evidence. Epidemiological studies are used to determine the prevalence and incidence of diseases, as well as the effectiveness of interventions. Economic evaluation (EE) studies are used to estimate the efficiency of interventions in improving population health, i.e., the cost of the intervention required to achieve a health improvement [1]. EEs can also provide the expected budget impact of offering a new intervention. This information can help policymakers prioritize which interventions to offer [2,3].

In high-resource countries, independent health technology assessment (HTA) agencies assess the availability and relevance of evidence to inform decision-makers on which health interventions to offer to the population, how to price them, and how to reimburse them [4]. HTA agencies aim to ensure that public healthcare systems achieve their fundamental objectives of not only using resources in the most efficient and effective way, i.e., producing as much health as possible with the resources invested in the healthcare system, but of also answering social preoccupations such as equity or social justice [5,6]. The province of Quebec has its own HTA agency, functioning independently from the national HYA agency. However, there may be collaborations aimed at ensuring a certain equity in terms of services offered to the population across the country.

In the field of neonatology, one of the main questions facing HTA agencies is that of the health conditions to be included in a screening program, whether pre- or post-natal care is included. Answering these questions can be particularly difficult, given the rarity of many congenital conditions that are candidates for a screening program [7]. The rarity of conditions can indeed restrict the ability to conduct statistically significant epidemiological studies [8,9]. Additionally, performing Cost-Utility Analysis (CUA) studies with non-verbal children can be challenging. However, one solution is to involve proxies, such as parents or healthcare professionals, to provide insights into the children’s quality of life [10]. Proxy respondents are typically tasked with completing standardized Health-Related Quality of Life (HRQoL) questionnaires on behalf of their child [11]. This process can be challenging, as it involves making subjective judgments about a child’s quality of life, which may not fully reflect the child’s experience [12,13]. The validity of CUA and Quality of Life (QoL) studies used to judge the value of adding a condition into a screening program can therefore be debatable [14,15]. Moreover, one should also ideally consider the value that society places on the life of a child, even if they have a disability or chronic condition, that might differ from the value of adults [16,17]. This child value can lead to pressure from various stakeholders, including parents, industry, and healthcare professionals, to provide rapid access to innovations, even if these interventions are poorly evaluated. The strong societal value placed on children can be a powerful motivator. Interest groups can use it to emphasize the potential benefits of interventions for young children. From a scientific point of view, it remains important to ensure these interventions have undergone a thorough evaluation before being offered in a public health care system [18]. These groups may push for quick access to new interventions, even when these have not been thoroughly evaluated. This pressure can create challenges in ensuring that decisions are based on sound scientific evidence and that the benefits and risks of different interventions are carefully weighed before making decisions regarding the offer of a new intervention in the health care system.

In the field of EE, CUAs have a specific advantage. They enable the use of a health outcome indicator, the quality-adjusted life year (QALY), which can be applied to any intervention. CUAs therefore make it possible to compare the cost/effectiveness of different interventions, and thus to judge the relative value of each one. CUAs are therefore of particular interest to decision-makers. As stated above, conducting CUA for interventions targeting children under 5 presents distinct challenges compared to those designed for older children and adults. However, the ability to conduct studies in this age group has recently been facilitated by the development of three measurement instruments, the Infant health-related Quality of life Instrument [19] (IQI), the Health Utilities Preschool [20] (HuPs), and the Toddler and Infant (TANDI) instrument. IQI, HuPs, and TANDI are measure instruments developed for assessing the quality of life in children under 5 years old [21,22]. They allow producing a value score that can be used in CUA studies. These three instruments have been validated and found to be reliable for their respective target populations [19,20]. As with any instrument, there is a need for further validation in diverse populations.

The IQI was developed at the University of Groningen [23]. It includes 7 health attributes: sleeping, feeding, breathing, stooling/poo, mood, skin, and interaction. The IQI is administered through a mobile application and is designed to assess overall HRQoL in infants up to 1 year of age [19,23]. HuPs is based on the Health Utilities Index (HUI) and is recommended for children aged 2–5 years. It was developed in Canada and Australia. It includes 12 items: emotion, hearing, speech, ambulation, dexterity, learning and remembering, thinking and problem-solving, pain, behavior, general health, self-care, and vision [20]. The TANDI instrument is another tool developed for assessing the quality of life in very young children, specifically those aged 1–36 months [21]. The TANDI includes six dimensions with three levels of report and general health measured on a Visual Analogue Scale from 0 to 100 [21].

Despite the advancements in CUA research, certain issues remain unresolved. In particular, the choice of dimensions still requires the use of proxies. These new instruments, however, concentrate more on elements that parents and health professionals can observe.

The ability to perform more convincing CUA studies in children under 5 years old is a significant advantage for researchers. The new instruments might support the production of EE studies in the field of neonatal screening. Consequently, policymakers can expect to be better equipped to ensure that chosen interventions are not just economically sound but also enhance children’s overall well-being. In a context where there is increasing pressure to incorporate epidemiological and economic data into the decision-making process for public services, the expectation for CUA studies in children under 5 years old remains an open question. This study aimed to explore this aspect.

## 2. Materials and Methods

### 2.1. Design

A qualitative case study was conducted to explore how policymakers perceive the role of EEs in the decision-making of an HTA committee reviewing interventions for children under 5, in light of the methodological innovations that allow to carry out cost-effectiveness analyses in this population. 

The participants of the study were voting members of an HTA agency’s scientific committee or members of the Ministry of Health’s neonatal screening advisory committee. They were solicited to participate in the study due to their expertise in health interventions for children under 5 years old.

### 2.2. Conceptual Framework

The study used the neo-institutional theory (NIT) as its conceptual framework [24,25] (Figure 1). NIT is a useful framework for understanding how institutionalized factors, such as norms, practices, and systems, can constrain the consideration of scientific evidence in decision-making. In this study, we focused on three main dimensions of NIT: institutional rules and norms, power relations, and interpretive schemes [25,26].

Institutional rules and norms are the formal and informal regulations, guidelines, and standards that govern behavior within a specific context. They provide a framework for understanding and guiding individual and organizational behavior, shaping expectations, defining appropriate practices, and influencing decision-making processes [24,25,27,28]. Rules and norms can be formal or informal. Formal rules and norms are explicitly stated and codified, while informal rules and norms are unwritten and socially shared. Formal rules are typically established by authoritative bodies, such as governments, regulatory agencies, or professional associations. They are often written, enforceable by law, and have a clear legal status. Informal rules and norms emerge through social interactions, cultural values, and collective practices. 

Power relations refer to the distribution and exercise of power among actors within a given institutional context. Power can be exerted through various means, including control of resources, authority, and influence. Power dynamics shape the interactions and decision-making processes within institutions, influencing which actors and interests have the most influence and the outcomes that emerge [24,25,29]. 

Interpretive schemes are the cognitive frameworks, meanings, and cultural understandings through which individuals and organizations interpret and make sense of their social world. They shape perceptions, beliefs, values, and actions, guiding behavior and influencing the adoption or resistance toward certain institutional rules, norms, and practices [24,25,29,30]. 

These three factors are expected to influence the balance of types of data available to policymakers through the funding given to researchers and the importance given by HTA agency committee members to different types of research.

This framework was expected to allow a deeper understanding of the institutional factors that influence the use of epidemiological and economic evidence in the decision-making process regarding the offering of health interventions for children under the age of 5.

### 2.3. Data Collection

A convenience sample of key policymakers with experience in the policy-making process for neonatal healthcare in the province of Quebec, either as a member of a scientific committee of the provincial health technology assessment agency or as a member of an advisory committee on newborn screening programs of the Ministry of Health, was recruited.

Semi-structured interviews were conducted using an interview guide developed based on the conceptual framework. The interview guide contained the following theme of the discussion: What factors reduce arbitrariness in choosing diseases for postnatal screening in Quebec? What structural mechanisms influence these decisions? What societal values impact these decisions? Are these elements effective? How do economic evaluations reduce arbitrariness? What else could be done to minimize arbitrary decisions? Does the field of neonatal screening bring some specificity to the place of economic evaluation in the decision-making process? It allows the exploration of the policymakers’ perceptions of the use of EEs in neonatal healthcare, in particular, regarding neonatal screening programs, their experiences in using EEs in decision-making, and the institutional factors that influence the use of EEs and the perspective regarding the place of EEs in the decision-making process, considering the possibility to nowadays conduct CUA on a population of under 5 years of age. The interviews were audio-recorded and verbatim transcribed.

Data collection continued until data saturation was reached. Thematic analysis was used to analyze the collected data, which involved identifying patterns and themes within the data. The analysis was conducted using an Excel spreadsheet.

In addition to the in-depth interviews, document analysis was conducted. Documents related to policies and reports regarding the use of evidence in neonatal healthcare in the Province of Quebec were looked for. Relevant documents were obtained both from the Internet and directly from some respondents. They included guidelines, reports, and other relevant publications. Top of Form.

### 2.4. Data Analysis

The data were analyzed using thematic analysis (Table 1). This involved identifying, coding, and organizing themes and patterns within the data. The coding process involved systematically categorizing the data into relevant themes and subthemes, based on the conceptual framework and emerging themes from the data. Two researchers (VHH and DR) coded the data independently. Any disagreements were resolved through discussion and consensus-building. If there had been disagreements, the third researcher would have been involved in discussions to resolve them.

The analysis was an iterative process, with the data being constantly reviewed and questions refined considering emerging themes and patterns. 

### 2.5. Validity of Results

The validity of the results is based on the following strategies [31]: Credibility was ensured through the triangulation of data sources. Transferability was enhanced through the description of the context and participants. Reliability was promoted through the use of a research protocol that was revised by a scientific committee and the involvement of three researchers. Confirmability was maintained through the documentation of the research process and by sending a first version of the analysis to three randomly selected interviewed persons, who were asked to evaluate whether the main messages they conveyed during the interviews were presented in the document. All three interviewees said they felt their opinions were represented in the results.

### 2.6. Ethical Considerations

The project was approved by the Laval University Research Ethics Committee number: 2021-253 A-1 R-1/07-06-2023. 

## 3. Results

Out of 13 individuals invited, 10 agreed to participate in an interview. Among these participants, seven are voting members of a HTA agency’s scientific committee and three are clinicians and members of the Ministry of Health’s neonatal screening advisory committee. Four participants are women. Four are actively involved in neonatal screening programs. 

In the following section, the results of the analysis will be presented by dimensions of the conceptual framework: structure, power relations, and interpretative schemes.

### 3.1. Structure

In the province of Quebec, the institutionalized factors that form the foundation of the neonatal care decision-making process are fundamentally rooted in the provincial HTA agency’s law. This law establishes a two-tiered structure: first, a scientific committee of the HTA agency makes a recommendation regarding a new condition to the offer of services in the public health care system; second, after consulting the Advisory committee on newborn screening, the Minister of Health makes the final decision on whether to follow the HTA agency’s recommendation. The HTA agency is mandated by law to conduct a precise evaluation of scientific evidence regarding the value of the proposed intervention. The law also requires the HTA agency to ensure that diverse opinions on this matter are incorporated into the final report. This is intended to minimize arbitrary decision-making. The challenge is then to be able to capture the diversity of opinions of the groups concerned, in particular patients, while the scientific committees are made up of health professionals.

Although the Ministry of Health can request that the HTA agency prioritize the evaluation of a specific intervention, it cannot interfere with the evaluation process itself. Therefore, the risk of arbitrary decision-making lies at the level of the final ministerial decision. At this level, factors beyond public health considerations, such as economic and political factors and scientific evidence may come into play, often in closed-door discussions. However, all respondents agree that it is extremely rare for the final decision not to follow the HTA agency’s recommendations.

“*… while the HTA’s role is to provide recommendations based on scientific evidence, the final decision lies with the minister. The minister typically follows the HTA’s recommendations, but there can be exceptions*.” (P9)

At the HTA agency level, evidence-based decision-making is the primary driver of its recommendations. The interviews highlighted the importance of HTA agency committee members in making recommendations based on robust scientific evidence, particularly epidemiological research. This is not only to justify the investments of taxpayer funds but also to uphold the legitimacy of evidence from high-quality epidemiological studies and EEs. Making decisions based on scientific evidence helps to prevent harm and promote the common good. Not basing decisions on this evidence would be an ethical problem, as it would go against the ethical principle of non-maleficence, which requires avoiding harm and minimizing risks to individuals and society as a whole. Not basing decisions on this evidence would also put committees at risk of folding under the influence of lobbyists who promote treatments for which evidence is weak.

“*Working in neonatal care can be challenging due to political pressures exerted by various groups, including parents and advocacy groups. These pressures can complicate decision-making processes and make it difficult for healthcare professionals to respond effectively to demands from politicians who may not fully grasp the practical realities of their work…*” (P6)

One notes that none of the respondents raised the question of the feasibility or challenges of conducting CUA in neonatology. Instead, the primary focus of their discourse focused on distinct issues such as the rarity of diseases in this context and the unique status that children hold within the social landscape. Children are the most valued group of the population. Almost all respondents consider that this value must be taken into account when the decision on the offer of innovation has to be made, even if numerical data showing differences in social value between sub-groups of the population are not available. Being able to have economic studies conducted on children under 5 was seen as an asset in the fight against arbitrary decisions or resistance to economic or emotional pressures but does not reduce the importance given to social and ethical issues.

“*…economic evaluations are aligned with other evaluations and factors, and decisions to cover expensive treatments may have already been made before an economic evaluation is conducted…*” (P1)

### 3.2. Power Relations

All respondents acknowledged the challenge of ensuring that all members of scientific committees have an equal opportunity to express their views. Committees include representatives from the medical, paramedical, and public sectors, and make efforts to give each group a voice. However, the potential conflicts of interest arise from gifts and benefits offered to doctors by pharmaceutical companies, the complexity of integrating new doctors into approval committees, the diversity of backgrounds among clinicians in committees, and the inherent biases and presuppositions that can influence decisions.

The second challenge faced by committees is the lack of expertise in pediatrics when discussing interventions for young children. Experts are typically consulted during the preliminary stages of the evaluation process when the literature is reviewed to identify relevant data. These data are then summarized in a report for the committee members to review. However, this expertise is often missing during the actual debates, as the committees are not typically made up of pediatric experts. As a result, the specificity of neonatal care may not be adequately considered in the evaluation process. 

“*…the importance of diverse expertise in committee debates and the challenges that can arise when specific expertise, such as pediatric expertise, is lacking…*”(P5)

There is a risk that the deliberations will mirror the way discussions are held for interventions for adults and that the lack of solid data in the epidemiological and economic literature will be given undue weight. Additionally, there is a risk that committee members may be influenced by external factors, such as political pressure or pharmaceutical companies. These external influences could affect not only the selection of diseases or treatments for neonatal screening programs but also the entire decision-making process, sometimes unconsciously. There is a consensus among participants that power dynamics beyond the committee’s control can shape decisions and potentially compromise the objectivity of the process.

A final concern expressed by most respondents is that the issue of power extends beyond social hierarchies to the influence of individual personalities. Individuals who are articulate and persuasive can sway outcomes, potentially leading to biased decisions. Respondents acknowledge that it is difficult to control this aspect through rules. A process that requires all committee members to express their opinions and vote on a new intervention can only mitigate the influence of certain members’ personalities, but it cannot eliminate it. 

“*…There are some members who are more vocal and persuasive than others, and they can influence the outcome of the decision… however if everyone expresses their opinions during the discussion "le tour de table", and how this allows for a more informed and fair decision-making process*”(P7)

The mere presence of instruments for conducting CUA studies in children under 5 years old does not necessarily guarantee a complete salutation to these concerns, although it does provide more satisfactory tools for such studies to be conducted. Consequently, some respondents fear that these economic studies from the healthcare perspective could give more weight to quantitative data in the decision-making process.

“*In my experience, even though we have tools for conducting Cost-Utility Analysis (CUA) studies in children under 5 years old, it doesn’t necessarily resolve all concerns related to this age group. These tools certainly enhance our ability to conduct such studies, but they don’t provide a complete solution. There are various factors at play and the mere existence of these tools doesn’t guarantee a comprehensive resolution*.”(P1)

### 3.3. Interpretive Schemes

Respondents agreed that the process for assessing scientific evidence is adequate. All recognized that the use of a highly standardized procedure to evaluate interventions, the importance given to epidemiological and economic evidence, and the consideration of other factors, such as social and ethical considerations, are all important safeguards for ensuring that decisions are made on a scientific basis and are socially legitimate. There is also general agreement that the process is flexible, which is seen as necessary when dealing with interventions for young children, as solid data are often lacking.

“*…they mainly focus on evaluating scientific data in their role on decision-making committees but also acknowledge the importance of considering other factors, such as ethical and social considerations …*” (P8)

However, participants who are directly involved in providing laboratory and clinical services to affected children are concerned that the current evaluation process does not give them enough opportunity to influence the decisions. These participants are typically members of expert advisory committees, but they are not voting members of the HTA agency scientific committees. They can provide input at the beginning of the evaluation process, and comment on the final report, but they feel that they have little influence on the overall decision-making process. According to these participants, the current situation means that experiential knowledge and practical knowledge about the organization of health services are not typically incorporated into final recommendations. This puts those who provide care for children at risk of experiencing undesirable consequences of final decisions, particularly about their workload. While these participants are concerned with offering interventions that would improve the well-being of children, they believe that defining which intervention can improve well-being cannot be reduced to a question of effectiveness, efficiency, and budgetary impact. This divergence between voting members of the HTA agency and members of expert advisory committees is also evident in the role of EEs in decision-making about innovations in neonatology. The committee members rely on HTA agency methodologists to assess the relevance of economic studies conducted on the interventions under review. For these respondents, the ability to conduct CUA studies in children under 5 years of age would strengthen the ability of the agency’s committees to fulfill their mandate to promote evidence-based interventions.

“*CUA studies would help us understand not just the effectiveness of an intervention, but also its impact on resources. This would put us in a much stronger position to recommend truly evidence-based approaches that benefit both children and the healthcare system as a whole*.” (P10)

As for those who offer services directly to children, they say little about the question of methodological advances. Their concern relates more particularly to their ability to increase their level of influence on the interventions to be evaluated, the analysis of existing data, and the drafting of recommendations.

In short, the interviews carried out show a broad agreement between the three elements of the conceptual framework, suggesting that the process of evaluating interventions for children under 5 years old is satisfactory and therefore unlikely to change significantly, even if the implementation of economic evaluations in this age group is less and less questionable.

## 4. Discussion

Two main findings emerge from the data. First, there is a consensus that the approach to assessing the value of neonatal interventions, which relies on an HTA agency, is satisfactory for minimizing arbitrariness in the recommendations made regarding the provision of interventions in the healthcare system. Second, decision-makers do not envisage that the possibility of now carrying out cost-utility evaluations for children under the age of 5, would lead the decision-making process to put the same emphasis on epidemiological and economic evidence as for interventions aimed at an older population than under 5 years-old children. 

The first finding is based on the fact that HTA agencies are conducted using a systematic approach to gather and assess evidence on the clinical effectiveness, cost-effectiveness, and other relevant aspects of health interventions. This approach helps to ensure that recommendations are made fairly and transparently and that they are based on the best available evidence.

Yet, respondents acknowledged the challenges and ethical considerations involved in decision-making about interventions for young children. They believe that a specific approach is needed. Interestingly, this position was shared by all respondents, regardless of their discipline, role in deliberating committees, or level of expertise in childcare. 

The study also puts in evidence the desire of policymakers to see the value of experiential knowledge in a highly specialized field more acknowledged in the decision-making process, especially when the literature is limited. Considering the satisfaction with the current process led by the HTA agency, it emerged that the expertise in neonatal screening is best served in a structure where this expertise is centralized in expert advisory committees, rather than in deliberative committees that make recommendations regarding the provision of technologies in the health system. This structure is seen as the most effective way to take into consideration the experts’ perspectives if it is involved in the decision-making process at the onset of the evaluation process [32] and the critical step between the recommendation made by the HTA agency and the decision taken by the minister. 

One notes that the minimization of arbitrariness is also satisfactory to participants because it is promoted by the fact that committees have a dynamic that forces members to respect the diversity of opinions. The predominant influence of professionals is mitigated by the key role of ethicists and the desire to present the voice of the representatives of the population in the reports. This dynamic is seen by all as being essential for a system that can accept the idea that, when dealing with young children, the key importance of epidemiological and economic evidence must be put into perspective. Ethicists, playing a crucial role within these committees, ensure that ethical considerations are thoroughly examined when assessing neonatal interventions [33]. Ethical consideration gives particular importance to the voice of the beneficiaries and the check-up for any risk of arbitrariness. This ethical position becomes especially important when making decisions that impact vulnerable populations, such as infants.

The second finding is that respondents are confident that the possibility of now carrying out CUA in young children should not have a major impact on the flexibility desired for the evaluation of health intervention for young people. This claim is largely based on the power structure of the decision-making process, which minimizes the influence of those with primarily economic interests in promoting interventions. Industry representatives and clinicians who take care of concerned sick children have little opportunity to influence the debates of the deliberative committees that review the reports produced by the HTA agency technical team. If members of a deliberative committee have clinical activity in neonatology, they are outnumbered by colleagues whose main professional interest is focused on other populations. 

Finally, the structure of the HTA agency includes mechanisms to address potential conflicts of interest. Experts in specific domains may have personal or professional interests to defend when participating in the evaluation of healthcare interventions, so the risk of bias or undue influence is acknowledged. The rules and guidelines are in place to identify potential conflicts of interest of committee members and minimize the influence on the decision-making process. Pressures by some groups, such as politicians, may be unavoidable, but their impact is limited to influencing the prioritization of health intervention evaluations. In any case, the emphasis on transparency and accountability throughout the evaluation process helps to maintain public trust in the HTA agency’s recommendations [34,35,36].

In summary, decision-makers welcome the efforts of researchers to produce more relevant economic studies with children under 5 years of age. They believe that the current structure of decision-making in neonatal health policies is satisfactory and that the new avenues for research will not affect the importance of being given to the specificity of young children when making decisions about the services they should receive.

## 5. Conclusions

Structural factors, interpretative schemes, and power management between groups concur to ensure that providing services to newborns in public health care is not hindered by the difficulty of producing evidence. It also ensures that the decision-making process considers the specificity of neonatology, particularly the social value given to children not captured by available evidence.

## Figures and Tables

**Figure 1 IJNS-10-00036-f001:**
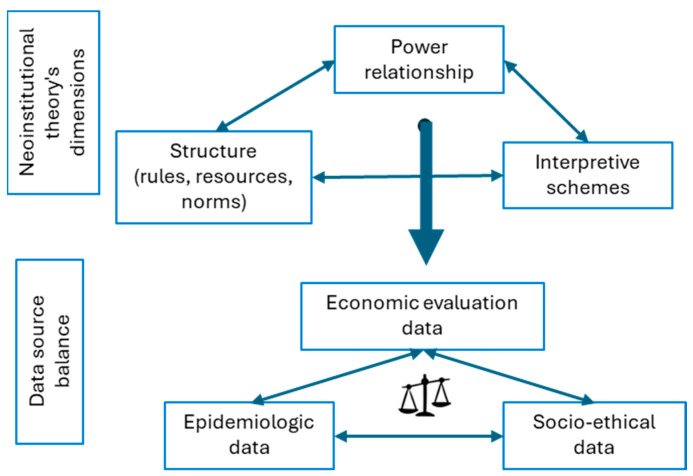
Neo-institutional theory.

**Table 1 IJNS-10-00036-t001:** Thematic analysis.

Main Themes	Subthemes
Institutional Rules and Norms	Formal Rules and Norms
	Informal Rules and Norms
Power Relations	Control of Resources
	Authority
	Influence
Interpretive Schemes	Cognitive Frameworks
	Cultural Understandings

## Data Availability

The data presented in this study are available on request from the corresponding author due to ethical reasons.

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
