# Peer review of "Economic Rationality in Decision-Making Regarding Newborn Screening: A Case Study in Quebec"

_2409-515X, 2024, doi:10.3390/ijns10020036_

Round 1

Reviewer 1 Report

Comments and Suggestions for Authors

The paper leaves many questions in my mind (see below) and insufficient attention appears to have been paid to details - e.g. the font size is inconsistent in places, CUA is defined in two places, EE is not defined, the figure is not described or referenced in the text.  The points below need to be addressed:

CUA defined in line 65 and again in line 88. Please elaborate briefly on the difficulty of CUA (i.e. is there a standardized questionnaire somewhere defining the element to be included?

I am unsure about the authors’ intended use of the abbreviation EE (line 152, 153, 154).  Does it  mean ‘economic evaluation’ (line 32,35,85, 139…) or maybe ‘economic evidence’ (line 43, 47 …). Please clarify. 

Please give a reference for ISPOR healthcare intervention guidelines (line76) and HTAi good practice guidelines (line 77).

Please give a reference to recommendations that other factors … beyond health benefits are more important (lines 80-82).

Please give references to the two groups developing IQI and HuPS (lines 91-92)

What is the qualitative case study referenced in line 108?  Please briefly describe. What role/influence/experience did each interviewee have in the case study? Was the study appropriate for the persons interviewed?  Need to know something about the case study.

Please provide the Interview Guide as supplemental information.

No reference to Figure 1 appears in the text. It could be explained a bit – for example, how does the 4-ccornered intersection work or should it not be viewed as an intersection? Why is the use of economic evaluation on the right of the decision-making process and not the left?

What role, if any, does system-wide preparedness play in the decision making process.  Are there differences in whether the analytical system is blood or urine, for example?  Any consideration given to other similar questions/considerations already addressed in other provinces?

Is there any role for national harmonization (equity) in the decision-making process or is Quebec a standalone Province governed totally by its own HTA-A? Please comment somewhere near the beginning.

Comments on the Quality of English Language

Data are plural - 'these' data instead of 'this' data

Reviewer 2 Report

Comments and Suggestions for Authors

The manuscript reports an interesting qualitative study of perceptions of policymakers with experience in the policy-making process for either neonatal healthcare or newborn screening in the province of Quebec regarding the use of economic evaluations in decision-making. The authors have clearly described the study design, conceptual framework, and data collection. However, the process of data analysis, including the methodology employed to derive the study findings (themes and subthemes), requires further elucidation. Although the study findings appear plausible and should be of interest, the documentation does not meet current standards for qualitative research. In the Methods section, the authors noted that data underwent thematic analysis, resulting in the identification of themes and subthemes. We suggest incorporating a table or chart (thematic synthesis) to present these subthemes and themes, as is standard practice in qualitative research. In addition, the manuscript could benefit from the inclusion of verbatim quotes from interviews to illustrate the identified sub-themes and themes and would enhance transparency. Several guidelines or checklists are available for reporting qualitative research. We encourage researchers to visit the Equator Network website as well as similar platforms and choose a tool to report their findings.  

On the other hand, the information contained in the Introduction and Discussion sections is inadequately documented and does not accurately characterize methods and applications of health economic evaluations. A manuscript that addresses the economic evaluation of pediatric interventions should reflect a solid understanding of the relevant literature and contain appropriate citations to that literature.  The discussion reiterates findings already presented in the results and do not provide any context to the published literature on either HTA processes or the role of economic evaluations in HTAs.  

Specific comments

L30-32: This sentence conflates scientific evidence with economic evidence, which are two distinct types of evidence. Scientific evidence refers to the effectiveness of healthcare intervention in improving clinical outcomes. Economic evidence refers to the balance of improved health outcomes with costs incurred, as the remainder of the paragraph acknowledges.

L33-34: Epidemiological data on prevalence and incidence of disease are only one kind of data that are used in assessing the potential benefits of interventions. Other crucial components include the health and quality of life of patients, which can differ by the types of economic evaluation, as well as cost-related information.

L42-43: The assertion, “The fact that HTA agencies give the utmost importance to epidemiological and economic evidence”, appears to be contradicted by reference 8, which emphasizes the importance of societal and patient values and perspectives in HTA. O’Rourke and colleagues recently responded (PMID: 35787309) to a comment on their paper that they characterized as based on “narrow and outdated understanding of HTA methods, processes, and practices.” That characterization appears to apply to the present manuscript.

L52-54: This refers to the traditional or “narrow and outdated understanding of HTA” referred to by O’Rourke and colleagues.

L55-56: It is incorrect and misleading to attribute the focus of clinical research on clinical effectiveness to what “is important for healthcare professionals.” Medical research is not funded by healthcare professionals. It is public and private funders and those who provide the sources of funding (elected politicians and investors) who decide on the objectives of medical research.

L57-61: It is misleading to equate neonatology with rare diseases. Neonatology in large part focuses on perinatal outcomes that are not rare, and conversely research on rare diseases is across the lifespan, not primarily during the neonatal period. Yes, it is harder to conduct scientific research on rare diseases, but that is not specific to neonatology. References 13 and 14 are not about neonatology or neonatal healthcare but rather are specific to newborn screening and genetic testing.

L61-63: It is true that decision making on neonatal care is the responsibility of clinicians and parents. However, that has nothing to do with the fact that babies cannot speak since that is true for pediatric care in general.

L63-65: This statement reflects incomplete understanding of cost-utility analyses (CUA). CUAs do not necessarily depend on quality-of-life questionnaires as stated. Multiple methods are used to general preference-based health-related quality of life (HRQoL) or health state utility values that are used to estimate expected changes in quality-adjusted life-years (QALYs) which in turn are used to quantify outcomes in CUAs. Quality-of-live questionnaires per se cannot be used to calculate QALYs.

L68-70. Although it is true that various stakeholders, including parent advocacy organizations and commercial enterprises often influence government decisions, including promoting coverage of less effective drugs in federal or provincial health plans, that is not specific to pediatric interventions. The widespread adoption and retention of low-value healthcare treatments and procedures is a major challenge for healthcare systems and society.

L65-68: Numerous economic studies have sought to quantify the relative value of lives lived during childhood vs adulthood using either stated preferences or revealed preferences, but that literature is not mentioned; reference 17 refers to an entirely different concept of the “value of children.”

L74-75: It is illogical to assert that “specific challenges in the field of neonatology” explain the need for guidelines for economic evaluations in pediatric care in general.  

L76-77: Why are there no citations to the two guidelines that are discussed? If one pastes “Good Practices for Health Technology Assessment of Pediatric Interventions” in a search engine, nothing turns up.

L78-84: We reserve comment on these statements until proper documentation is provided.

L85-88: We strongly disagree with the assertion that CUAs of interventions targeted to children under 5 are “impossible”; CUAs for interventions targeted to infants and young children are often conducted and reported. CUAs have employed preference-based HRQoL measures originally developed for either adult populations or older children. In addition, the authors offer no evidence to support their contention that economic evaluations play less of a role for interventions for younger children than for older children.

L89-92: The authors mention two newly developed QOL measures for children under 5, IQI and Health Utilities Preschool-HuPs. However, they fail to provide supporting information, including citations. Which groups developed the instruments? What are their properties? Have they been independently validated? The simple existence of measures does not mean that they have validity and can be considered useful for CUAs. It is essential to assess the reliability and validity of newly developed instruments for young children with a range of illnesses and disabilities, including neurodevelopmental disabilities. The lack of psychometric evidence for preference-based HRQoL instruments in children with disabilities is a significant concern. PMC7065222.

L92-93: It is unclear how or why the authors think that these two quality-of-life measures can be used to assess life expectancy. Quality and quantify of life are different dimensions of health, which are combined in order to estimate QALYs.

L93-94: Even if instruments are validated, they cannot be used to generate QALYs used in CUAs until researchers use those instruments to collect data on children affected by interventions.

L103: It is unclear what exactly the authors mean by “the societal value of children” or how that would be expected to be incorporated in decision-making processes.

L169-170: If there were disagreements, was the third researcher involved in discussions to resolve them?

L172-173: Reviewed by who? Two researchers (VHH and DR)?

L182-185: The authors mentioned that a first version of the analysis was sent to three randomly selected interviewees, and they were asked to evaluate whether the main messages they conveyed during the interviews were accurately represented. However, upon reading the results and discussion, we could not locate the outcomes of the evaluation.

L193: What is the meaning of HTA-A? That abbreviation was never defined. It appears to be a clumsy circumlocution to avoid using the name of the provincial HTA agency that was acknowledged at L147. The authors of the Quebec Ministry of Health and Social Services or use the agency’s name, INESSS.    

L272-274: The meaning of this sentence is unclear. We know that the authors believe that having preference-based HRQoL instruments for children under 5 is essential for the “feasibility” of CUAs of interventions targeting that age group, a belief which we have addressed in comments on L85-88. However, it is unclear that the simple availability of instruments will affect the number of CUAs. Presumably by “power dynamics” the authors meant that an increased number of CUAs for pediatric interventions could have some impact on policy decisions.

L274-276:  The meaning of this sentence is unclear. Economic evaluations in health can be done from a healthcare perspective, in which only medical costs and outcomes are considered, or from a societal perspective, in which a broader range of costs and outcomes are addressed. It would be helpful to know if the respondents were thinking of those two different types of economic evaluation studies or were instead worried that all economic evaluations are focused on medical costs and outcomes.  

L312: Here the authors for the first time use the indefinite article before HTA-A, from which it can be deduced that the term was meant to refer to a generic HTA agency.

L328-330: These two sentences and the cited reference appear to refer to research on child subjects rather than to the question of how to assess decisions on paying for evidence-based services for young children, which is the topic of the manuscript.

L338-341: References 27 and 28 discuss NBS recommendations made by clinical experts in the United States. Since neither has anything to do with the role of expert opinion in HTA or health policy processes, they do not belong.

L342-343: Reference 29 and 30 describe patient care programs, not health care policy deliberative processes; appropriate references must be cited.

L344: The correct adjective is “neonatal”.  

L359-362: Neither reference 32 nor 33 provide any support for these statements. Reference 32 refers to research on pediatric subjects, not healthcare. Reference 33 is a classic economic article about how the increased cost of childbearing as societies become richer leads families to choose to have fewer children; it too has nothing to do with healthcare of the societal value of children.

L362-365: It is unclear what “limitations” are alluded to. It is also not clear that references 34-36 have any connection to policymakers’ deliberations about interventions for young children.

Round 2

Reviewer 1 Report

Comments and Suggestions for Authors

Thank you to the authors for succinctly their responses to this reviewer's questions and suggestions.  Aside from a typo in L-44 (HTA instead of HYA), I am satisfied with the revised manuscript.

Author Response

Thank you, reviewer, for all of your comments that helped improve the manuscript.

Reviewer 2 Report

Comments and Suggestions for Authors

General comments

The authors addressed many specific comments on the writing and the citation of references. However, while we appreciate the progress that was made, our major concerns were not addressed. In particular, the authors did not acknowledge or respond to the two paragraphs of overarching comments.

“The manuscript reports an interesting qualitative study of perceptions of policymakers with experience in the policy-making process for either neonatal healthcare or newborn screening in the province of Quebec regarding the use of economic evaluations in decision-making. The authors have clearly described the study design, conceptual framework, and data collection. However, the process of data analysis, including the methodology employed to derive the study findings (themes and subthemes), requires further elucidation. Although the study findings appear plausible and should be of interest, the documentation does not meet current standards for qualitative research. In the Methods section, the authors noted that data underwent thematic analysis, resulting in the identification of themes and subthemes. We suggest incorporating a table or chart (thematic synthesis) to present these subthemes and themes, as is standard practice in qualitative research. In addition, the manuscript could benefit from the inclusion of verbatim quotes from interviews to illustrate the identified sub-themes and themes and would enhance transparency. Several guidelines or checklists are available for reporting qualitative research. We encourage researchers to visit the Equator Network website as well as similar platforms and choose a tool to report their findings.”  

“On the other hand, the information contained in the Introduction and Discussion sections is inadequately documented and does not accurately characterize methods and applications of health economic evaluations. A manuscript that addresses the economic evaluation of pediatric interventions should reflect a solid understanding of the relevant literature and contain appropriate citations to that literature.  The discussion reiterates findings presented in the results and does not provide appropriate context to the published literature pertaining to either HTA processes or the role of economic evaluations in HTAs.”

Among the authors’ responses to our specific comments that were indexed to line numbers in their original submission, several were either insufficient or missed the point of the comment.

For example, the authors do not appear to have understood our comment, “CUAs do not necessarily depend on quality-of-life questionnaires as stated. Multiple methods are used to general preference-based health-related quality of life (HRQoL) or health state utility values that are used to estimate expected changes in quality-adjusted life-years (QALYs) which in turn are used to quantify outcomes in CUAs. Quality-of-life questionnaires per se cannot be used to calculate QALYs.” Although the authors state that they agree with these comments, their revised text is not consistent with that stance. See specific comments below.  

Although the authors acknowledge that they had cited an inappropriate reference in support of the statement about the relative value of years lived during childhood and adulthood and state that they “have modified the reference for this sentence”, the two new references appear no more pertinent than the previous reference. Reference 13, which was published in an obscure journal, “examines the associations between six indicators of childhood socioeconomic status with classic late-adulthood health outcomes.” Reference 14 examined old data from the Republic of Korea from an epoch in which parents viewed children as economic investments, which is no longer the case, to model a life-cycle optimization problem of the optimal childbearing strategy. Neither reference has nothing to do with the relative values placed on the lives and health of children and adults. The authors should either familiarize themselves with the relevant economic literature or delete the assertion.    

Also, although the authors were responsive in citing references for the IQI and HuPS quality of life measures that they had previously suggested as being suitable for use in children under 5 years of age and explaining where and how those two measures were developed, their responses raise additional questions that must be addressed.  

The authors acknowledge that the two measures were developed for different age groups, the IQI for infants under 12 months of age and the HuPS for children ages 2-5 years. That begs the question of how analysts could be expected to use either or both measures children from birth to age 5. Essentially the same text appears in the authors’ previously published article in JIEMS, which has the same limitations as the present submission. The authors fail to acknowledge that the IQI and HuPS measures are based on different frameworks for assessing utility. Consequently, it may not be suitable to combined the two measures in a single study. The literature has documented major differences in HRQL values between the EQ-5D and HUI frameworks, leading to often inconsistent estimates of QALY losses or gains associated with different conditions and treatments.  

The authors’ focus on the IQI and HuPS may be misplaced. The authors cite reference 18 as supporting the validity of the IQI or HuPS, which is incorrect since neither measure was mentioned. Instead, reference 18 provides information on a third instrument, the TANDI, which was initially tested using mothers of infants and toddlers ages 1-36 months. A 2021 article from the same group reported that the TANDI instrument is also appropriate for children ages 3-4 years. It appears that the TANDI, unlike the IQI or HuPS, is potentially appropriate for assessing HRQL in children from infancy through age 4 years. 

The authors were not sufficiently responsive to the following comment: “The simple existence of measures does not mean that they have validity and can be considered useful for CUAs. It is essential to assess the reliability and validity of newly developed instruments for young children with a range of illnesses and disabilities, including neurodevelopmental disabilities. The lack of psychometric evidence for preference-based HRQoL instruments in children with disabilities is a significant concern. PMC7065222.”

Although the authors cited studies in which the developers of the two instruments reported information on validity, that is not sufficient for the instruments to be regarded as ready for use in CUAs. Additional research studies need to be undertaken, as explained in the previous comment, before these or other instruments can be considered useful for CUAs in newborn screening. For example, Mao et al. (Vaccine 2023) cite IQI and TANDI (with no mention of HuPS) as "candidates for future utility estimation" but evidently did not consider the measures suitable for current practice.

The authors misinterpreted the following comment in their response.

“L93-94: Even if instruments are validated, they cannot be used to generate QALYs used in CUAs until researchers use those instruments to collect data on children affected by interventions.” “The reviewer raises an important point: the target population can differ from the general population. We cannot be sure that the validation fully applies to sub-groups of the population. The paragraph was rewritten accordingly.”

Although the authors addressed the comment about lack of validation in population subgroups, they did not address the comment that CUAs can only use data that has already been collected. Until researchers have fielded instruments such as the TANDI in infants and young children with NBS-related conditions, it is unclear how the existence of those instruments would have any relevance for the preparation of CUAs in NBS.

Specific comments on the revised manuscript

L33-34: The clause, “the investment cost needed to produce a health improvement, thanks to the intervention under evaluation” is unclear. We suspect that the authors meant, “the cost of the intervention needed to produce a health improvement”, which would make sense. 

L50-51: It is true that when a condition is rare, there may be limitations in terms of the available sample size for a study. Small sample sizes can create difficulties in obtaining statistically significant results. However, many epidemiologic studies on rare diseases have been published. More important challenges in assessing the epidemiology of rare diseases in newborn screening include the difficulty of finding appropriate comparison groups for newborn screening cohorts owing to ascertainment and selection biases in clinical case series. If the authors wish to address the epidemiologic literature on newborn screening, they should cite pertinent references.

L52-54: The authors assert that “it may also prove difficult to conduct economic studies because of the difficulty of carrying out Cost-Utility Analysis (CUA) and Quality of Life (QoL) studies with children they are unable to speak.” This statement has multiple problems.

·       It is misleading to assert that economic “studies” of newborn screening are difficult to carry out because of challenges in conducting QoL studies in infants and young children.

·       Many CUAs of NBS have been published. For example, several published CUAs on NBS for severe combined immunodeficiency (SCID) used expert opinion on the frequencies of health states (severe, moderate, or mild) together with expert judgments on the loss of utility associated with each of those states.

·       Not all economic evaluations are CUAs; in fact, the majority are not. See the review article by Png et al. PMID: 36272385.

·       Most QoL studies do not provide information that can be used to calculate QALYs for CUAs.

·       QoL studies are not required for CUAs. CUAs can use either QALY or DALY metrics (see Png et al.) Both QALY and DALY estimates are frequently calculated without conducting QoL studies.

·       QALYs are valued using either indirect preference-based health-related quality of life (HRQoL) measures or direct utility elicitation methods such as Time Trade-Off (TTO) or Standard Gamble to generate health utilities.

·       Patient samples are not required. The Canadian guidelines for health economic evaluations call for researchers to use population preferences rather than patient preferences for calculating health utilities.

·       QALYs or DALYs are calculated over the entire lifespan, not just infancy and early childhood. The bulk of QALY gains or averted DALYs from NBS occurs after age 5 years

·       A more important barrier to conducting CUAs for newborn screening may be the lack of appropriate estimates of health state utilities for people living with rare disorders targeted by NBS regardless of age.

L55-57: The statement that proxy respondents “are tasked not only with answering Utility and QoL questionnaires instead of their children but also with defining the dimensions of QoL concepts used to produce QoL and utility scores and determining their levels for a specific child” is incorrect. There are no 'Utility' questionnaires and parents are not asked to define the dimensions of QoL concepts. To the contrary, proxy respondents are asked to fill out standardized HRQoL questionnaires associated with preference-based instruments such as EQ-5D, HUI3/HUI2, and CHU-9D. We encourage authors to refer to PMCID: PMC293474 for a better understanding of preference-based HRQoL instruments and how to estimate health utility values.

L66-70: These three sentences accurately characterize the potential impact of parent narratives on decision makers, but the terminology appears judgmental. Different wording might be helpful. Also, reference 15 does not appear to be pertinent.

L76: We suggest that the authors substitute “measure” for “indicator”.  

Comments on the Quality of English Language

See specific comments. 

Author Response

Response to Reviewer 2:

General comments

The authors addressed many specific comments on the writing and the citation of references. However, while we appreciate the progress that was made, our major concerns were not addressed. In particular, the authors did not acknowledge or respond to the two paragraphs of overarching comments.

“The manuscript reports an interesting qualitative study of perceptions of policymakers with experience in the policy-making process for either neonatal healthcare or newborn screening in the province of Quebec regarding the use of economic evaluations in decision-making. The authors have clearly described the study design, conceptual framework, and data collection. However, the process of data analysis, including the methodology employed to derive the study findings (themes and subthemes), requires further elucidation. Although the study findings appear plausible and should be of interest, the documentation does not meet current standards for qualitative research. In the Methods section, the authors noted that data underwent thematic analysis, resulting in the identification of themes and subthemes. We suggest incorporating a table or chart (thematic synthesis) to present these subthemes and themes, as is standard practice in qualitative research. In addition, the manuscript could benefit from the inclusion of verbatim quotes from interviews to illustrate the identified sub-themes and themes and would enhance transparency. Several guidelines or checklists are available for reporting qualitative research. We encourage researchers to visit the Equator Network website as well as similar platforms and choose a tool to report their findings.” 

We appreciate the suggestions made by the reviewer. We have therefore added a table presenting the themes and subthemes of the analysis, as well as the verbatim quotes in the result section.

On the other hand, the information contained in the Introduction and Discussion sections is inadequately documented and does not accurately characterize methods and applications of health economic evaluations. A manuscript that addresses the economic evaluation of pediatric interventions should reflect a solid understanding of the relevant literature and contain appropriate citations to that literature.  The discussion reiterates findings presented in the results and does not provide appropriate context to the published literature pertaining to either HTA processes or the role of economic evaluations in HTAs.”

Among the authors’ responses to our specific comments that were indexed to line numbers in their original submission, several were either insufficient or missed the point of the comment.

For example, the authors do not appear to have understood our comment, “CUAs do not necessarily depend on quality-of-life questionnaires as stated. Multiple methods are used to general preference-based health-related quality of life (HRQoL) or health state utility values that are used to estimate expected changes in quality-adjusted life-years (QALYs) which in turn are used to quantify outcomes in CUAs. Quality-of-life questionnaires per se cannot be used to calculate QALYs.” Although the authors state that they agree with these comments, their revised text is not consistent with that stance. See specific comments below. 

Although the authors acknowledge that they had cited an inappropriate reference in support of the statement about the relative value of years lived during childhood and adulthood and state that they “have modified the reference for this sentence”, the two new references appear no more pertinent than the previous reference. Reference 13, which was published in an obscure journal, “examines the associations between six indicators of childhood socioeconomic status with classic late-adulthood health outcomes.” Reference 14 examined old data from the Republic of Korea from an epoch in which parents viewed children as economic investments, which is no longer the case, to model a life-cycle optimization problem of the optimal childbearing strategy. Neither reference has nothing to do with the relative values placed on the lives and health of children and adults. The authors should either familiarize themselves with the relevant economic literature or delete the assertion.   

L57 We are sorry for this mistake, we have updated the new reference (12,13) for this sentence.

Also, although the authors were responsive in citing references for the IQI and HuPS quality of life measures that they had previously suggested as being suitable for use in children under 5 years of age and explaining where and how those two measures were developed, their responses raise additional questions that must be addressed

The authors acknowledge that the two measures were developed for different age groups, the IQI for infants under 12 months of age and the HuPS for children ages 2-5 years. That begs the question of how analysts could be expected to use either or both measures children from birth to age 5. Essentially the same text appears in the authors’ previously published article in JIEMS, which has the same limitations as the present submission. The authors fail to acknowledge that the IQI and HuPS measures are based on different frameworks for assessing utility. Consequently, it may not be suitable to combined the two measures in a single study. The literature has documented major differences in HRQL values between the EQ-5D and HUI frameworks, leading to often inconsistent estimates of QALY losses or gains associated with different conditions and treatments

The reviewer is right. We understand your concern about the use of the two measures, IQI and HuPS, in the same study given they were developed for different age groups and are based on different frameworks for assessing utility.

In this manuscript, we do not suggest using both measures in the same study. Instead, we acknowledge the existence of these two measures and their respective age groups. Our intention was to highlight the different tools available for different age groups, not to suggest that they should be used interchangeably or combined in a single study.

We agree that combining two measures based on different frameworks in a single study may lead to inconsistent estimates of QALY losses or gains. This is a valid point and we appreciate you bringing it to our attention. 

The authors’ focus on the IQI and HuPS may be misplaced. The authors cite reference 18 as supporting the validity of the IQI or HuPS, which is incorrect since neither measure was mentioned. Instead, reference 18 provides information on a third instrument, the TANDI, which was initially tested using mothers of infants and toddlers ages 1-36 months. A 2021 article from the same group reported that the TANDI instrument is also appropriate for children ages 3-4 years. It appears that the TANDI, unlike the IQI or HuPS, is potentially appropriate for assessing HRQL in children from infancy through age 4 years.

We appreciate your suggestion to consider the TANDI instrument, which seems to cover a broader age range from infancy through age 4 years. We agree that it is crucial to use the most appropriate measure for assessing Health-Related Quality of Life (HRQL) in children. We have modified the paragraph and its references. We apologise for any confusion caused.

L79-83 “…the ability to conduct studies in this age group has recently been facilitated by the development of three measurement instruments, the Infant health-related Quality of life Instrument (IQI), the Health Utilities Preschool (HuPs) and the Toddler and Infant (TANDI) instrument. IQI, HuPs, and TANDI are measure instruments developed for assessing the quality of life in children under 5 years old…”

The authors were not sufficiently responsive to the following comment: “The simple existence of measures does not mean that they have validity and can be considered useful for CUAs. It is essential to assess the reliability and validity of newly developed instruments for young children with a range of illnesses and disabilities, including neurodevelopmental disabilities. The lack of psychometric evidence for preference-based HRQoL instruments in children with disabilities is a significant concern. PMC7065222.”

We appreciate the comment made by the reviewer. We acknowledge that for every instrument, there is indeed a need for validation in diverse populations, as the validity assessed in a specific population may not necessarily apply to others. We understand that the mere existence of measures does not guarantee their validity or usefulness for CUAs. It is indeed essential to assess the reliability and validity of newly developed instruments for young children with a range of illnesses and disabilities, including neurodevelopmental disabilities.

We, therefore, added a sentence in the manuscript emphasizing the need for further validation studies in diverse populations: L85 “As with any instrument, there is a need for further validation in diverse populations”

Although the authors cited studies in which the developers of the two instruments reported information on validity, that is not sufficient for the instruments to be regarded as ready for use in CUAs. Additional research studies need to be undertaken, as explained in the previous comment, before these or other instruments can be considered useful for CUAs in newborn screening. For example, Mao e t al. (Vaccine 2023) cite IQI and TANDI (with no mention of HuPS) as "candidates for future utility estimation" but evidently did not consider the measures suitable for current practice.

We apologise for any confusion caused by our previous manuscript. We have revised the manuscript and references to focus on these three instruments, IQI, HuPs and TANDI. We acknowledge that while these instruments have reported valid information, this alone may not be sufficient for them to be considered ready for use in Cost-Utility Analyses (CUAs). As the reviewer rightly pointed out, additional research studies need to be undertaken before these or other instruments can be deemed useful for CUAs in newborn screening. This study focuses more on the potential changes in the decision-making process.

The authors misinterpreted the following comment in their response.

L93-94: Even if instruments are validated, they cannot be used to generate QALYs used in CUAs until researchers use those instruments to collect data on children affected by interventions.”

“The reviewer raises an important point: the target population can differ from the general population. We cannot be sure that the validation fully applies to sub-groups of the population. The paragraph was rewritten accordingly.”

Although the authors addressed the comment about lack of validation in population subgroups, they did not address the comment that CUAs can only use data that has already been collected. Until researchers have fielded instruments such as the TANDI in infants and young children with NBS-related conditions, it is unclear how the existence of those instruments would have any relevance for the preparation of CUAs in NBS.

The reviewer raises a very difficult point: data can be difficult to obtain, even if utilities and QALYs can be produced, due essentially to the fact that many screenable diseases are rare diseases. Simulations with experts’ opinion input are feasible, but not fully satisfactory. We rewrote the paragraph to highlight this issue.

L98 “Despite the advancements in CUA research, certain issues remain unresolved. In particular, the choice of dimensions still requires the use of proxies. These new instruments, however, concentrate more on elements that parents and health professionals can observe.”

Specific comments on the revised manuscript

L33-34: The clause, “the investment cost needed to produce a health improvement, thanks to the intervention under evaluation” is unclear. We suspect that the authors meant, “the cost of the intervention needed to produce a health improvement”, which would make sense.

We agree that the sentence was unclear, therefore, we have modified the clause: L33 “the cost of the intervention required to achieve a health improvement”.

L50-51: It is true that when a condition is rare, there may be limitations in terms of the available sample size for a study. Small sample sizes can create difficulties in obtaining statistically significant results. However, many epidemiologic studies on rare diseases have been published. More important challenges in assessing the epidemiology of rare diseases in newborn screening include the difficulty of finding appropriate comparison groups for newborn screening cohorts owing to ascertainment and selection biases in clinical case series. If the authors wish to address the epidemiologic literature on newborn screening, they should cite pertinent references.

Thank you for the suggestion, The references have been updated for the paragraph to be focused on the newborn screening program.

L52-54: The authors assert that “it may also prove difficult to conduct economic studies because of the difficulty of carrying out Cost-Utility Analysis (CUA) and Quality of Life (QoL) studies with children they are unable to speak.” This statement has multiple problems.

  • It is misleading to assert that economic “studies” of newborn screening are difficult to carry out because of challenges in conducting QoL studies in infants and young children.

  • Many CUAs of NBS have been publpublished CUAs on NBS for severe combined immunodeficiency (SCID) used expert opinion on the frequencies of health states (severe, moderate, or mild) together with expert judgments on the loss of utility associated with each of those states.

  • Not all economic evaluations are CUAs; in fact, the majority are not. See the review article by Png et al. PMID: 36272385.

  • Most QoL studies do not provide information that can be used to calculate QALYs for CUAs.

  • QoL studies are not required for CUAs. CUAs can use either QALY or DALY metrics (see Png et al.) Both QALY and DALY estimates are frequently calculated without conducting QoL studies.

  • QALYs are valued using either indirect preference-based health-related quality of life (HRQoL) measures or direct utility elicitation methods such as Time Trade-Off (TTO) or Standard Gamble to generate health utilities.

  • Patient samples are not required. The Canadian guidelines for health economic evaluations call for researchers to use population preferences rather than patient preferences for calculating health utilities.

  • QALYs or DALYs are calculated over the entire lifespan, not just infancy and early childhood. The bulk of QALY gains or averted DALYs from NBS occurs after age 5 years

  • A more important barrier to conducting CUAs for newborn screening may be the lack of appropriate estimates of health state utilities for people living with rare disorders targeted by NBS regardless of age.

 We agree with the comments made by the reviewer. The comments made us realise that our text was a little bit out-of-topic in addressing more than what was our main point, the capacity to conduct scientifically acceptable CUA studies. We modified the paragraph taking into account the reviewer’s comments:

L49-53 “The rarity of conditions can indeed restrict the ability to conduct statistically significant epidemiological studies. Additionally, performing Cost-Utility Analysis (CUA) studies with non-verbal children can be challenging. However, one solution is to involve proxies, such as parents or healthcare professionals, to provide insights into the children’s quality of life.”

L55-57: The statement that proxy respondents “are tasked not only with answering Utility and QoL questionnaires instead of their children but also with defining the dimensions of QoL concepts used to produce QoL and utility scores and determining their levels for a specific child” is incorrect. There are no 'Utility' questionnaires and parents are not asked to define the dimensions of QoL concepts. To the contrary, proxy respondents are asked to fill out standardized HRQoL questionnaires associated with preference-based instruments such as EQ-5D, HUI3/HUI2, and CHU-9D. We encourage authors to refer to PMCID: PMC293474 for a better understanding of preference-based HRQoL instruments and how to estimate health utility values.

We agree with the reviewer, the sentence was not correct, it has been revised as follows:

L53-55 “Proxy respondents are typically tasked with completing standardized Health-Related Quality of Life (HRQoL) questionnaires on behalf of their child.”

L66-70: These three sentences accurately characterize the potential impact of parent narratives on decision-makers, but the terminology appears judgmental. Different wording might be helpful. Also, reference 15 does not appear to be pertinent.

The three sentences were revised:

L63-67 “The strong societal value placed on children can be a powerful motivator. Interest groups can use this to emphasize the potential benefits of interventions for young children. From a scientific point of view, it remains important to ensure these interventions have undergone a thorough evaluation before being offered in a public health care system.”

L76: We suggest that the authors substitute “measure” for “indicator”. 

The word has been substituted to be “indicator”.

Round 3

Reviewer 2 Report

Comments and Suggestions for Authors

The authors were fully responsive.